# Comparison of Implant Placement Accuracy in Healed and Fresh Extraction Sockets between Static and Dynamic Computer-Assisted Implant Surgery Navigation Systems: A Model-Based Evaluation

**DOI:** 10.3390/ma15082806

**Published:** 2022-04-11

**Authors:** Miaozhen Wang, Xiaohui Rausch-Fan, Yalin Zhan, Huidan Shen, Feng Liu

**Affiliations:** 1First Clinical Division, Peking University School and Hospital of Stomatology & National Clinical Research Center for Oral Diseases & National Engineering Laboratory for Digital and Material Technology of Stomatology & Beijing Key Laboratory of Digital Stomatology, 37 A Xishiku Street, Xicheng District, Beijing 100034, China; wmiaozhen@126.com (M.W.); zhanyalin2014@126.com (Y.Z.); shen_hd@pku.edu.cn (H.S.); 2Division of Conservative Dentistry and Periodontology, Center of Clinical Research, Dental School, Medical University of Vienna, 1090 Vienna, Austria; xiaohui.rausch-fan@meduniwien.ac.at

**Keywords:** accuracy of implant placement, static computer-assisted implant surgery, dynamic computer-assisted implant surgery, fresh extraction socket

## Abstract

The aim of this model-base study was to compare the accuracy of implant placement between static and dynamic computer-assisted implant surgery (CAIS) systems in a fresh extraction socket and healed ridge. A randomized in vitro study was conducted. Twenty 3D-printed maxillary models and 80 implants were used. One experienced researcher placed the implants using either the static navigation or dynamic navigation system. Accuracy was measured by overlaying the real position in the postoperative CBCT on the virtual presurgical placement of the implant in a CBCT image. Descriptive and bivariate analyses of the data were performed. In the fresh sockets, the mean deviation was 1.24 ± 0.26 mm (entry point), 1.69 ± 0.34 mm (apical point), and 3.44 ± 1.06° (angle discrepancy) in the static CAIS group, and 0.60 ± 0.29 mm, 0.78 ± 0.33 mm, and 2.47 ± 1.09° in the dynamic CIAS group, respectively. In the healed ridge, the mean deviation was 1.09 ± 0.17 mm and 1.40 ± 0.30 mm, and 2.12 ± 1.11° in the static CAIS group, and 0.80 ± 0.29 mm, 0.98 ± 0.37 mm, and 1.69 ± 0.76° in the dynamic CIAS group, respectively. Compared with the static CAIS system, the dynamic CAIS system resulted in significantly lower entry and apical errors in both fresh sockets and healed ridges. Differences in bone morphology therefore seem to have little effect on accuracy in the dynamic CAIS group.

## 1. Introduction

Dental implant prostheses are an effective replacement for missing teeth [1]; however, inaccurate implant positioning can predispose patients to inferior outcomes as well as short- and long-term complications [2,3]. Use of cone-beam computed tomography (CBCT) allows for 3-dimensional (3D) jawbone images to be obtained with low doses of radiation [4], thereby facilitating the evaluation of bone structures and neighboring teeth [5].

Conventional methods of implant placement are often unable to reliably reproduce the optimal planned implant position at the surgical site. Computer-assisted implant surgery (CAIS) therefore utilizes CBCT combined with software in order to carry out virtual implant placement in the optimal 3-dimensional (3D) position [6]. Presurgical virtual 3D plans can then be transferred during actual surgical implant placement using static or dynamic CAIS [7].

The static CAIS system involves the use of a computer-aided design and manufactured template with a “sleeve” that precisely guides drilling and implant placement [8,9]. Thus, the static guide allows precise transfer of the planned implant position, while the dynamic CAIS system involves real-time tracking of implant drilling and the outline of the patient’s jaw in real-time using a navigation monitor. In doing so, it is therefore easy to identify any deviation of the drill or implant away from the planned position in real-time, allowing the surgeons to adjust the drilling depth or angle as well as the position of the implant as needed.

Both static and dynamic CAIS systems therefore offer greater accuracy when they transfer the virtually planned implant position to the surgical situation compared to conventional freehand placement [8,9,10,11,12]. However, deviation can also occur with CAIS systems due to errors in radiographic data acquisition, data processing, and guide-template manufacturing, and in accordance with the type of template used, tolerance of the guiding sleeve, levels of guidance during drilling and implant placement, and the registration procedures, not to mention human error [7,13], all of which could cause deviation from the virtual implant position. Osteotomy within sites presenting asymmetrical bone density can result in shifting of the drill toward the path of least resistance, particularly in fresh extraction sockets during conventional placement [10]. Despite this, however, few studies have compared the accuracy between static and dynamic CAIS systems, particularly during immediate implant placement in a fresh extraction socket for immediate implant placement. Therefore, the aim of this model study was to compare static and dynamic CAIS systems in terms of implant placement accuracy in healed bone and fresh extraction sockets.

## 2. Materials and Methods

### 2.1. Study Design

This randomized study compared the degree of deviation in implant placement using static versus dynamic CAIS systems in vitro. The CONSORT guidelines were followed throughout the study [14].

The sample size was calculated based on the following formula and previous studies:

This resulted in a sample size of 80 implants (40 implants per group).

An experienced surgeon placed all 80 implants (diameter 4.1 mm length 10 mm) in 20 partially edentulous maxillary arch models. The 3D-printed polymethylmethacrylate maxillary models were used in this study. Twenty implants were placed in healed bone using the static CAIS system and 20 with the dynamic CAIS system, respectively, while 20 were also placed in fresh extraction sockets using both systems, respectively, giving 40 implants per group. The stereolithographic resin models contained two missing adjacent teeth in both the anterior (tooth positions 21 and 22, ISO-3950) and posterior (tooth positions 15 and 16, ISO-3950) areas. Fresh extraction sockets were simulated at sites 21 and 22 (ISO-3950). The randomization process was performed by a professor using a software program that generates random permuted blocks.

### 2.2. Scanning Procedure

Fiducial markers were placed on the teeth of the maxillary denture (Figure 1). The model was then stabilized by placing silicone rubber over the surface of the teeth during the CBCT scan. Scanning features were as follows: peak voltage, 90 kV; beam current, 8.0 mA; scan time, 8.0 s; and field of view, 17 mm × 13.5 mm. The models were subsequently subjected to 3D intraoral surface scans. The CBCT imaging and DICOM data were then analyzed using 3D implant-planning software to generate the implant positions. An STL file associated with the intraoral scan was imported into the software and merged with the DICOM image. The CBCT data and DICOM data were then matched via alignment of the anatomic landmarks in the model of the upper jaw.

### 2.3. Implant Placement Plans

All of the implant placement plans for both groups were devised by a single operator who would also perform the model-based surgeries. Implants were planned virtually based on the prosthetic design and bony anatomy [3] using the merged CBCT data (.dicom) and oral scanning files (.stl) in 3Shape software.

In the static CAIS group, the templates were fabricated using the stereolithography technique (MED 610 and 705). A commercial T-sleeve (diameter 5 mm) was embedded in the CAIS system, and a drilling handle was used for both the pilot and twisted drills (Figure 2). In the dynamic CAIS group, data were directly imported into the dynamic navigation system.

### 2.4. Osteotomy and Implant Placement

All surgical procedures were carried out by the same experienced operator using conventional methods assisted by each CAIS system. All surgeries were performed in the same operation room with the same instruments within one week. Implants (4.1 mm × 10 mm) were inserted with the aid of a digital template in the static CAIS group and the dynamic navigation system in the dynamic CAIS group.

#### 2.4.1. Static CAIS Surgical Protocol

Osteotomy implants were prepared and placed in accordance with the guidelines provided with the implant system. The implants were placed using an implant carrier and burs (15 rpm) with a maximum torque of 50 N/cm.

#### 2.4.2. Dynamic CAIS Surgical Protocol

The dynamic navigation system contains an infrared tracking camera plus two tracking sensors. During CBCT scanning, one tracking sensor was firmly attached to the handpiece, while the other sensor was attached to the adjacent dentition using silicone rubber to stabilize and retain the sensors. The U-shape tube with fiducial markers was used to register the maxillary model using the tracking camera. After the registration process, the markers were removed. The motion of the drill in relation to the maxillary model was tracked and displayed on a screen showing the planned implant position on CBCT images.

Selection of the virtual drill was carried out using an existing database. Preparation of the osteotomy site preparation and placement of the implant placement were performed using the navigation system on the monitor. The implants were placed as described above using an implant carrier and burs (Figure 3).

### 2.5. Outcome Measures

The navigation system used in this study included pre-setup validated software, allowing for the evaluation of implant placement accuracy. The preoperative virtual surgical plan and the postoperative CBCT scans were overlapped based on marker positions, then, the position of each implant in the postoperative scan was determined (Figure 4). Any deviation existing between the actual and virtual planned implant position was calculated automatically using the software. Deviations from the virtual plan were determined as described below (Figure 5):
3D deviation of the implant platform: displacement (in mm) between planned and actual implant positions, measured in the center of the implant platform.3D deviation of the implant apex: displacement (in mm) between the planned and actual implant positions, measured in the center of the implant apex.Depth deviation (mm): difference in depth along the long axis of the implant.Lateral deviation (mm): a 2D difference in mesial/distal (y-axis) and buccal/lingual (x-axis) placement of the implant (disregarding depth deviation).Angular deviation (°): deviation between the planned and actual positions at the center axis.

**Figure 4 materials-15-02806-f004:**
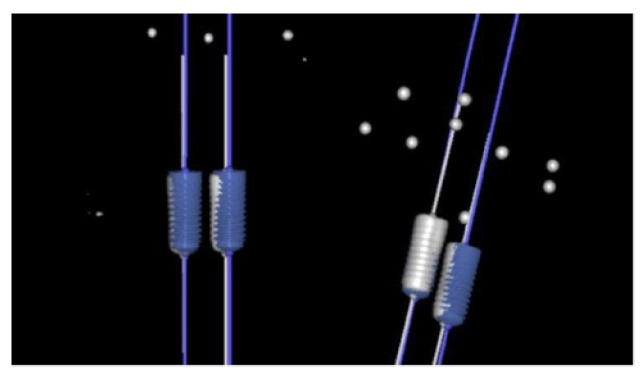
The overlapping of preoperative and postoperative 3D representations of the jaws allowed for the accuracy of implant placement to be evaluated because the preoperative representations of the planned implants were matched to their placed counterparts in the postoperative images.

**Figure 5 materials-15-02806-f005:**
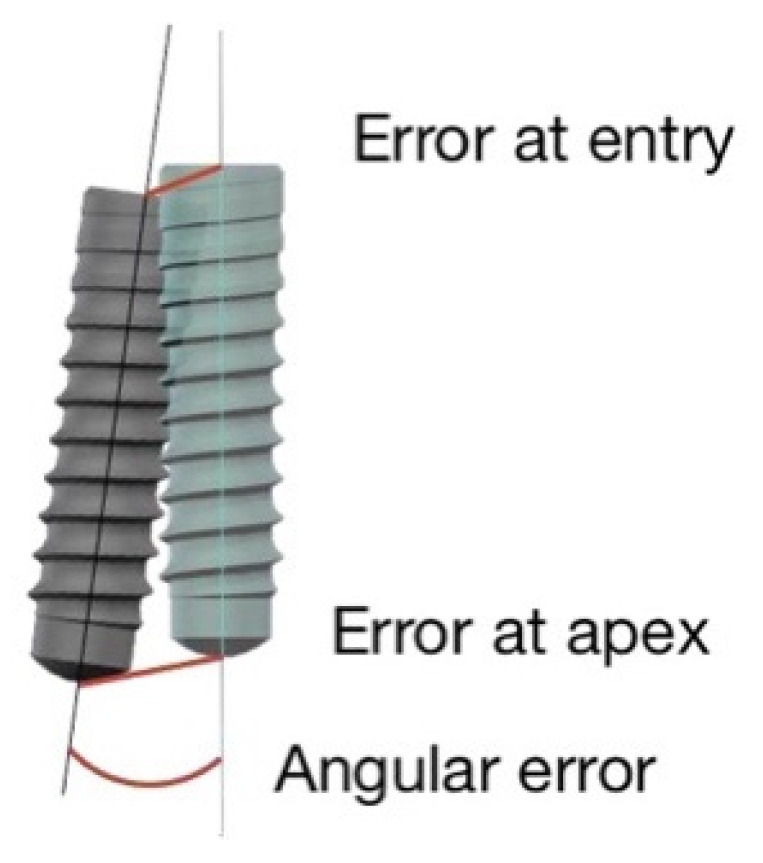
Parameters used to analyze the accuracy of the implant placement were error at entry, error at apex, and angular error. Error at entry is defined as 3D distance between the coronal center of the corresponding planned and placed implants. Error at apex is defined as the 3D distance between the apical center of the corresponding planned and placed implants. Angular deviation is calculated as the 3D angle between the longitudinal axes of the planned and placed implants.

The accuracy of the measurement was to the second decimal place. An additional outcome of interest was the surgical time in total between the initial drilling until completion of implant placement. The time required for registration was also recorded in the dynamic CAIS group.

The total surgery time from initial drilling until completion of implant placement was also determined, along with the time required for registration in the dynamic CAIS group. A second calibrated examiner blinded to the treatment plan and patient grouping carried out the accuracy evaluations. The final calculations were automated.

### 2.6. Statistical Analyses

Statistical analyses of all variables were carried out using SPSS software (ver. 22.0; IBM Corp., Armonk, NY, USA). Descriptive data were recorded as mean ± standard deviation (SD) for the quantitative variables, and the absolute numbers and percentages for the qualitative variables. The normality of data distribution was calculated using the Kolmogorov–Smirnov test. An independent two-sample *t* test was then applied to the data normally distributed. The Mann–Whitney U test was applied to all data showing non-normal distribution. Statistical significance was set as *p* < 0.05.

## 3. Results

The results were categorized by surgical site: anterior fresh extraction socket or posterior healed extraction socket.

With the anterior fresh extraction sockets, static CAIS resulted in mean implant position deviations at the platform and apex of 1.24 ± 0.26 and 1.69 ± 0.34 mm, respectively; while in the dynamic group, the mean deviations of 0.60 ± 0.29 and 0.78 ± 0.33 mm were observed, respectively (Table 1). Angular deviations of 3.44 ± 1.06 and 2.47 ± 1.09° were observed in the static and dynamic groups, respectively. Overall, statistically significant differences were observed between the two CAIS groups (*t*-test, *p* < 0.05). With the posterior healed extraction sockets, static CAIS resulted in mean implant position deviations at the platform and apex of 1.09 ± 0.17 and 1.40 ± 0.30 mm, respectively, while in the dynamic group, mean deviations of 0.80 ± 0.29 and 0.98 ± 0.37 mm were observed, respectively. Angular deviations of 2.12 ± 1.11° and 1.69 ± 0.76° were observed in the static and dynamic CAIS groups, respectively. Overall, statistically significant differences were found between the two CAIS groups (*t*-test *p* < 0.05, 0.001, =0.16) (Table 2).

Within the static CAIS group, the healed socket group showed higher implant accuracy in all parameters compared with the fresh socket group (*p* < 0.05) (Table 3). Meanwhile, within the dynamic CAIS group, the fresh socket group showed higher accuracy in terms of coronal and apical deviations compared with the healed socket group (*p* < 0.05) (Table 4). Lateral deviation in the anterior fresh extraction sockets was less in the dynamic CAIS group than the static CAIS group both in terms of buccal and distal placement of the implant at the implant platform and the implant apex (Figure 6).

With the fresh sockets, the mean drilling time was 15 min (range: 11–18 min) in the static CAIS group and 28 min (range: 20–35 min) in the dynamic CAIS group. Meanwhile, with the healed sockets, the mean drilling time was 14 min (range: 10–18 min) in the static CAIS group and 26 min (range: 18–34 min) in the dynamic CAIS group. The registration procedure in the dynamic CAIS group required an additional mean time of 10 min.

## 4. Discussion

Optimal 3D implant placement is essential during dental implant restorations, ensuring appropriate design as well as optimal functionality, satisfactory aesthetics, and peri-implant tissue health. The overall accuracy of implant placement has been greatly improved by the development of CAIS systems [8,9,10,11,12,15,16,17]. The risk of sinus perforation and inferior alveolar nerve injury during drilling have also been reduced by using these systems [10,17]. Minimal invasiveness during implant placement is also an important goal [18], particularly in high-risk patients such as those taking anticoagulation medications for cardiovascular issues and those with an edentulous, atrophic mandible.

In this study, both the static and dynamic CAIS systems allowed for accurate placement of a single implant in the healed posterior sockets, with mean platform and apex deviations between the planned and actual implant position of less than 1.1 and 1.5 mm, respectively, and a mean angular deviation of less than 3°. Recent systematic reviews revealed that the mean entry-point deviation between the planned and actual implant positions is 1.0 mm with static CAIS in cadavers and models, which is in agreement with our findings. Meanwhile, Chen et al. [19] reported mean angular, coronal, and apical deviations in cast models of 4.45 ± 1.97°, 1.07 ± 0.48 mm, and 1.35 ± 0.55 mm, respectively, using a dynamic navigation system. The mean coronal and apical deviations were similar to our findings, however, the angular deviation was lower in our study. In vitro studies published in 2015 [9] and 2019 [20] explored the accuracy of the dynamic navigation system and revealed similar findings to those observed here. The outcomes of the present study therefore seem to be in accordance with previous studies. However, the results should be interpreted with caution, since most previous analyses of the accuracy of dynamic systems have been carried out in vitro using artificial models, which have often shown superior findings compared with clinically obtained results [16]. In 2019, Dechawat Kaewsiri et al. [21] published a clinical RCT comparing the accuracy of two CAIS systems. The dynamic CAIS they used was IRIS. Patients who required a dental implant for at least three months were included in the study. The situation seems to be similar to our study for healed sites in vitro. In their study, mean platform and apex deviations between two groups were less than 1.05 mm and 1.29 mm, respectively, and mean apex deviation was less than 3.06°. They concluded that dynamic CAIS provided similar accuracy compared to static CAIS in a single tooth space. In our study, for the posterior healed sites, the static CAIS group showed similar accuracy compared to Dechawat Kaewsiri’s study, with mean platform and apex deviation less than 1.09 mm and 1.40 mm, respectively, and mean angular deviation less then 2.2°. The dynamic CAIS group showed more platform and apex accuracy compared to the static CAIS group for posterior healed sites, with mean platform and apex deviation less than 0.80 mm and 0.98 mm, respectively. Mean platform deviation had no significant different in two groups. The accuracy of CAIS was affected by many technical factors. The dynamic CAIS system we used was an active system, which means that the infrared light emitted from the surgical instrument and the patient were detected by infrared cameras. Compared to passive systems, active systems can limit the interference of tracking from nature light sources, which might slightly raise the accuracy to some extent. However, more research is needed to confirm this hypothesis.

Studies have shown that dynamic navigation systems perform as well as static systems [10,21], and significantly better than freehand implant placement [10,22]. A meta-analysis also revealed significantly higher implant placement accuracy at the entry point and apex when using dynamic navigation systems compared with traditional static systems. Various aspects of implant placement can result in errors such as inferior quality of CBCT scans, inaccuracies during registration or planning inaccuracy, a lack of precision during tracking, jaw movements, operator error in following the drilling path onscreen, and errors when overlaying the two sets of CBCT scans [5,16,23,24,25,26,27]. Poor-quality of the CBCT images can also affect the accuracy of the virtual planning procedures, leading to deviation between the resulting implant position and the planned position. Mean errors in image processing and segmentation of <0.5 mm have been reported, which although small, could still affect the final results [16,26,27,28,29,30].

Static CAIS systems facilitate drill and implant placement via a tooth-supported surgical guide, providing improved accuracy compared to bone- and mucosa-supported guides. Correct seating of the guide is of utmost importance, as even minor errors can be amplified during drilling of the osteotomy at the apex level. A systematic review by Bover-Ramos et al. [31] showed that fully-guided implant surgery resulted in greater accuracy than half-guided surgery. The implants in both groups were placed using fully guided surgery aimed to minimize errors.

The accuracy of dynamic CAIS systems is also affected by a number of factors. Target registration errors (TREs) refer to the deviation between points (other than fiducial points) on the CT image and the surgical site after registration. In our study, a non-invasive occlusal stent with fiducial markers was used for image registration, resulting in unacceptable accuracy. This implantation method is relatively easy to perform, especially for patients with sufficient teeth to support the tracking array. Using an occlusal stent, Luebbers et al. [32] found that the minimum TRE close to the maxillary teeth was 0.4 mm.

Additionally, human error can affect the accuracy, particularly when using a dynamic CAIS system. Without any mechanical guidance, the drills are controlled by the surgeon’s hand, depending on hand–eye-coordination and their ability to accurately interpret the images on the navigation monitor [7,9]. Our study employed one surgeon who had ample implant surgery experience (at least 500 implants inserted using conventional implantology, and at least 50 implants each using CAIS implantology). The surgeries were performed by the same surgeon so that the accuracy of the implant placement could be meaningfully compared between the groups.

Implant osteotomy in bone bed with asymmetric density and morphology could also result in deviation of the implant position, normally toward the sites with less resistance [10]. Few studies have investigated the utility of CAIS systems in guiding implant surgery at fresh extraction sockets. We found that the mean deviation at the entry point and apex was greater in fresh extraction sockets than in healed sites in the static group. One possible explanation for this finding is the presence of a gap due to the “rotational allowance” of the drill bit in the tube [33]. Although bone density was the same in both models, the implant was placed palatally, with less resistance on the labial side. Thus, the drill and implant might have slipped toward the labial side, which would be difficult to detect due to the poor static view offered by the CAIS system [34]. In the actual fresh extraction socket, the hard palatal bone wall would present resistance compared with the labial gap, which would make the labially slipped implant placement more possible. The deviation of the angular at the platform could predict the apical error. In the dynamic CAIS group, any influence of bone morphology on implant placement accuracy may be eliminated because the 3D position of the drill can be tracked and corrected in real time.

Dynamic navigation systems have several advantages over static systems [16,35]. For example, scanning, planning, and the actual implant surgery can be performed on the same day. In addition, the surgical plans can be adjusted during surgery when necessary, and the view of the surgical site is not obscured by the surgical guide. Other advantages include superior water cooling of the surgical site, and the fact that conventional implant instruments and drills can be used. On the other hand, the surgical time is typically longer when using a dynamic versus static CAIS system. Because dynamic systems provide no mechanical guidance, the surgeon needs more time for adjustments of the drill. Additional drawbacks of the dynamic CAIS include the requirement for a clear sight between the cameras and tracking sensors; the registration that must be performed prior to surgery; and the system is also expensive.

Studies that investigated the accuracy of guided systems could be broadly categorized as in vitro studies or clinical trials. In vitro studies are ideal for evaluating different systems as they control for many factors related to patients such as saliva, blood, and movement. Moreover, in vitro studies could control for a range of variables that limit imaging performance. Clinical trials have shown that the inability to identify the bone crest affects the depth prediction. Anatomic variability leads to greater deviation between the planned and actual implant depth, which often results in the underreporting of such deviation [16,17]. Although in vitro studies allow for the direct comparison of different systems, the nature of our particular in vitro study design limits the generalizability of the results.

This study showed that entry and apical errors, in both fresh and healed sockets, were significantly lower in the dynamic than static CAIS system. However, further research is needed to validate these results and determine whether there are any clinical complications associated with these navigation systems. Clinicians should be advised that greater apical deviation between the planned and actual implant positions may be expected when a static CAIS system is used for anterior fresh extraction sockets.

## 5. Conclusions

Within the limitations of this study, our results showed that compared to the static CAIS system, the dynamic CAIS system exhibited significantly lower entry and apical errors for both fresh sockets and healed sites. Differences in bone morphology exerted less influence on implant placement accuracy within the dynamic CAIS group. Clinical trials with a bigger sample size are necessary to determine the clinical utility of these computer-aided navigation procedures.

## Figures and Tables

**Figure 1 materials-15-02806-f001:**
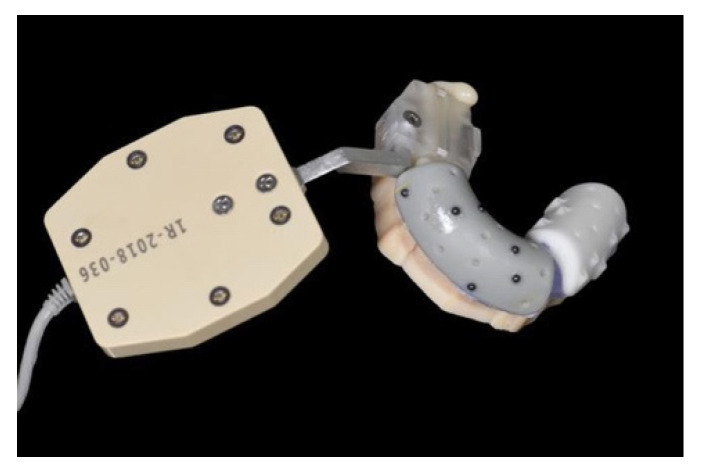
Maxillary model with registration template and fiducial markers.

**Figure 2 materials-15-02806-f002:**
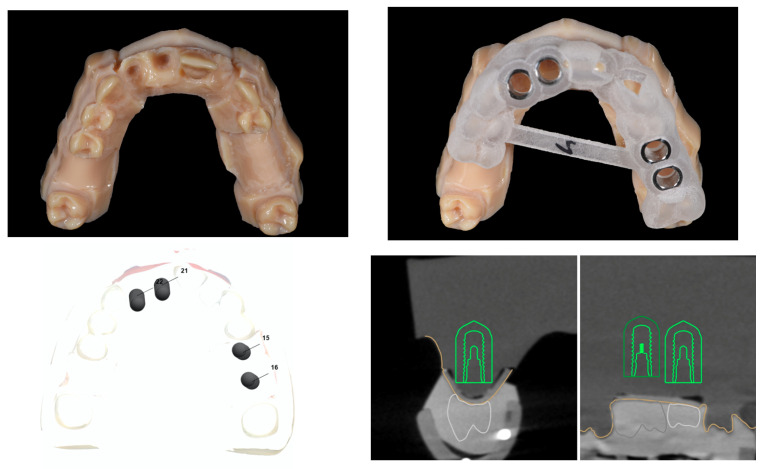
Dental implant planning with the Static CAIS system using a cone-beam computer tomography (CBCT) scan (green line). Virtual template design according to the planned virtual dental implant placement; manufactured stereolithography template fixed over the dental surface of the teeth and placed over the partially edentulous upper jaw models.

**Figure 3 materials-15-02806-f003:**
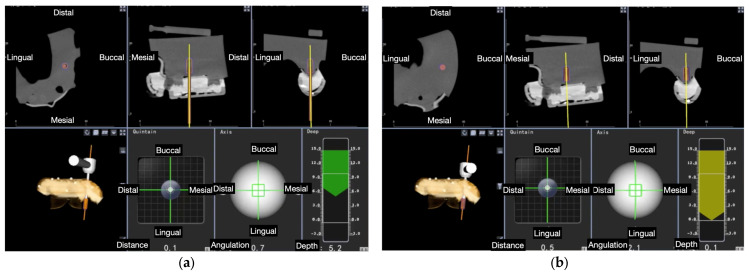
Dental implant placement controlled at all planes and depth: (**a**) Dental implant placement in process; (**b**) Dental implant placement achieved the planned depth.

**Figure 6 materials-15-02806-f006:**
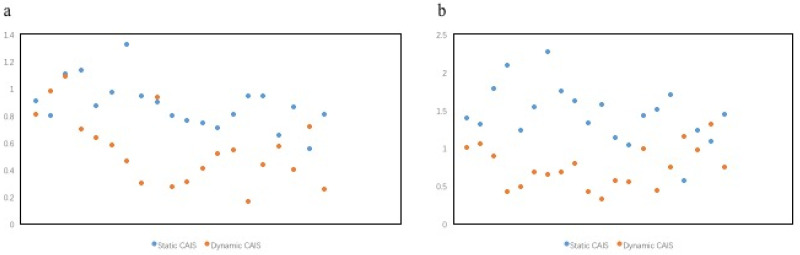
Lateral deviation in the anterior fresh extraction sockets in static CAIS and dynamic CAIS groups. (**a**) A 2-dimensional measure of the difference in buccal and distal placement of the implant at the implant platform. (**b**) A 2-dimensional measure of the difference in buccal and distal placement of the implant at the implant apex.

**Table 1 materials-15-02806-t001:** Descriptive deviation values at the apical (mm), coronal (mm), and angular (°) levels in the anterior sockets (sites 2.1 and 2.2).

		N	Mean	SD	*p*
CORONAL	Static	20	1.24	0.26	<0.001
	Dynamic	20	0.60	0.29	
APICAL	Static	20	1.69	0.34	<0.001
	Dynamic	20	0.78	0.33	
ANGULAR	Static	20	3.44	1.16	0.010
	Dynamic	20	2.47	1.09	

**Table 2 materials-15-02806-t002:** Descriptive deviation values at the apical (mm), coronal (mm), and angular (°) levels in the posterior region.

		N	Mean	SD	*p*
CORONAL	Static	20	1.09	0.17	<0.001
	Dynamic	20	0.80	0.29	
APICAL	Static	20	1.40	0.30	<0.001
	Dynamic	20	0.98	0.37	
ANGULAR	Static	20	2.12	1.11	0.161
	Dynamic	20	1.69	0.76	

**Table 3 materials-15-02806-t003:** Differences between the anterior area and posterior area in the static CAIS groups.

	N		Mean	SD	*p*
CORONAL	Anterior	20	1.24	0.26	0.044
	Posterior	20	1.09	0.17	
APICAL	Anterior	20	1.69	1.40	0.007
	Posterior	20	1.40	0.30	
ANGULAR	Anterior	20	3.44	1.16	0.001
	Posterior	20	2.12	1.11	

**Table 4 materials-15-02806-t004:** Differences between the anterior area and posterior area in the dynamic CAIS groups.

	N		Mean	SD	*p*
CORONAL	Anterior	20	0.60	0.29	0.034
	Posterior	20	0.80	0.29	
APICAL	Anterior	20	0.78	0.33	<0.001
	Posterior	20	0.98	0.37	
ANGULAR	Anterior	20	2.47	1.09	0.161
	Posterior	20	1.69	0.76

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
