# Peer review of "Comparison of Implant Placement Accuracy in Healed and Fresh Extraction Sockets between Static and Dynamic Computer-Assisted Implant Surgery Navigation Systems: A Model-Based Evaluation"

_materials, 2022, doi:10.3390/ma15082806_

Round 1

Reviewer 1 Report

The authors of the model-based studies analyze all aspects in detail.  However, the material of the model should be explained to prove its reliability and the degree to which it can simulate the bone.  Are there any other materials that are closer to the bone that can be substituted?  Or biomechanical materials can be applied, which can provide more confidence.  

The word "he" on line 71 should be changed to "the". In addition, all the figure numbers are not guided in this article and should be corrected.  The accuracy of the final measurement should be to the second decimal place and should also be stated in the article.

Author Response

Thank you very much for your kind comments. 

The material of the model was polymethylmethacrylate, which can simulate type II/III bone.

The word "he" on line 71 has been changed to "the".

The figure numbers have been guided and corrected in this article.

The accuracy of the final measurement have been stated in the article.

Reviewer 2 Report

This in vitro has a robust methodology with clear aims and objects comparing status vs dynamic CAIS systems in terms of accuracy in healed vs fresh extraction sockets. The sample size calculation was sound and based on previously published studies. Despite adhering to CONSORT guidelines, it wouldve been appropriate to mention the randomisation process that took place. The protocol appears to be sound with regards to execution of static vs dynamic implant placement, along side the outcome measures. It was also good to see data normality was tested apprioratly which resulted in all the data being collected to have non normal distribution.   Overall, a good research project that should be published in an area with very little evidence.

Author Response

Thank you very much for your kind comments. 

The randomization process was performed by a professor using a software programme that generates random permuted blocks. A corresponding description has been added in Materials and Methods.

Reviewer 3 Report

The manuscript "Comparison of implant placement accuracy in healed and fresh extraction sockets between static and dynamic computer-assisted implant surgery navigation systems : A model-based evaluation" shows that the dynamic computer‐assisted implant surgery system used to place implants resulted in significantly lower entry and apical errors in both fresh sockets and healed ridges. The paper is complete, well organized and adds informations on implant placement techniques in order to reduce the margin of error. However, further studies will be needed to validate this technique. the paper is accepted.

Author Response

Thank you very much for your kind comments.